# Constraining Linear-chain CRFs to Regular Languages

**Sean Papay, Roman Klinger, & Sebastian Padó**
University of Stuttgart
`(sean.papay|klinger|pado)@ims.uni-stuttgart.de`

## Abstract

A major challenge in structured prediction is to represent the interdependencies within output structures. When outputs are structured as sequences, linear-chain conditional random fields (CRFs) are a widely used model class which can learn *local* dependencies in the output. However, the CRF's Markov assumption makes it impossible for CRFs to represent distributions with *nonlocal* dependencies, and standard CRFs are unable to respect nonlocal constraints of the data (such as global arity constraints on output labels). We present a generalization of CRFs that can enforce a broad class of constraints, including nonlocal ones, by specifying the space of possible output structures as a regular language $\mathcal{L}$. The resulting regular-constrained CRF (RegCCRF) has the same formal properties as a standard CRF, but assigns zero probability to all label sequences not in $\mathcal{L}$. Notably, RegC-CRFs can incorporate their constraints during training, while related models only enforce constraints during decoding. We prove that constrained training is never worse than constrained decoding, and show empirically that it can be substantially better in practice. Finally, we demonstrate a practical benefit on downstream tasks by incorporating a RegCCRF into a deep neural model for semantic role labeling, exceeding state-of-the-art results on a standard dataset.

## 1 Introduction

Structured prediction is a field of machine learning where outputs are expected to obey some pre-defined discrete structure. Instances of structured prediction with various output structures occur in many applications, including computer vision (e.g., scene graph generation (Johnson et al., 2015) with graph-structured output), natural language processing (e.g., linguistic parsing (Niculae et al., 2018) with tree-structured output, relation extraction (Roth & Yih, 2004) with tuple-structured output) or modeling the spatial structure of physical entities and processes (Jiang, 2020).

A key difficulty faced by all models is to tractably model interdependencies between different parts of the output. As output spaces tend to be combinatorially large, special techniques, approximations, and independence assumptions must be used to work with the associated probability distributions. Considerable research has investigated specific structures for which such approaches make machine learning tractable. For instance, when outputs are trees over a fixed set of nodes, maximal arborescence algorithms allow for exact inference (Chu, 1965; Edmonds, 1967); when outputs are graph-structured, loopy belief propagation can provide approximate inference (Murphy et al., 1999).

If the output forms a linear sequence, a Markov assumption is often made: model outputs depend only on their immediate neighbors, but not (directly) on more distant ones. A common model that uses this assumption is the linear-chain conditional random field (CRF) (Lafferty et al., 2001), which has found ubiquitous use for sequence labeling tasks, including part-of-speech tagging (Gimpel et al., 2011) and named entity recognition (Lample et al., 2016). This model became popular with the use of hand-crafted feature vectors in the 2000s, and is nowadays commonly used as an output layer in neural networks to encourage the learning of structural properties of the output sequence. The Markov assumption makes training tractable, but also limits the CRF's expressive power, which can hamper performance, especially for long sequences (Scheible et al., 2016). Semi-Markov CRFs (Sarawagi & Cohen, 2004) and skip-chain CRFs (Sutton & McCallum, 2004) are techniques for relaxing the Markov assumption, but both come with drawbacks in performance and expressiveness.

In this work, we propose a new method to tractably relax the Markov assumption in CRFs. Specifically, we show how to constrain the output of a CRF to *a regular language*, such that the resulting *regular-constrained CRF (RegCCRF)* is guaranteed to output label sequences from that language. Since regular languages can encode long-distance dependencies between the symbols in their strings, RegCCRFs provide a simple model for structured prediction guaranteed to respect these constraints. The domain knowledge specifying these constraints is defined via regular expressions, a straightforward, well understood formalism. We show that our method is distinct from approaches that enforce constraints at decoding time, and that it better approximates the true data distribution. We evaluate our model empirically as the output layer of a neural network and attain state-of-the-art performance for semantic role labeling (Weischedel et al., 2011; Pradhan et al., 2012). Our PyTorch RegCCRF implementation can be used as a drop-in replacement for standard CRFs.

## 2 RELATED WORK

We identify three areas of structured prediction that are relevant to our work: constrained decoding, which can enforce output constraints at decoding time, techniques for weakening the Markov assumption of CRFs to learn long-distance dependencies, and weight-learning in finite-state transducers.

**Constrained decoding.**    A common approach to enforcing constraints in model output is *constrained decoding*: Models are trained in a standard fashion, and decoding ensures that the model output satisfies the constraints. This almost always corresponds to finding or approximating a version of the model's distribution conditionalized on the output obeying the specified constraints. This approach is useful if constraints are not available at training time, such as in the interactive information extraction task of Kristjansson et al. (2004). They present *constrained conditional random fields*, which can enforce that particular tokens are or are not assigned particular labels (positive and negative constraints, respectively). Formally, our work is a strict generalization of this approach, as position-wise constraints can be formulated as a regular language, but regular languages go beyond position-wise constraints. Other studies treat decoding with constraints as a search problem, searching for the most probable decoding path which satisfies all constraints. For example, He et al. (2017) train a neural network to predict token-wise output probabilities for semantic role labeling following the BIO label-alphabet (Ramshaw & Marcus, 1999), and then use exact A* search to ensure that the output forms a valid BIO sequence and that particular task-specific constraints are satisfied. For autoregressive models, constrained beam search (Hokamp & Liu, 2017; Anderson et al., 2017; Hasler et al., 2018) can enforce regular-language constraints during search. We further discuss constrained decoding as it relates to RegCCRFs in Section 5.

**Markov relaxations.**    While our approach can relax the Markov assumption of CRFs through non-local hard constraints, another strand of work has developed models which can directly *learn* nonlocal dependencies in CRFs: *Semi-Markov CRFs* (Sarawagi & Cohen, 2004) relax the Markov property to the semi-Markov property. In this setting, CRFs are tasked with segmentation, where individual segments may depend only on their immediate neighbors, but model behavior within a particular segment need not be Markovian. As such, semi-Markov CRFs are capable of capturing nonlocal dependencies between output variables, but only to a range of one segment and inside of a segment. *Skip-chain CRFs* (Sutton & McCallum, 2004) change the expressiveness of CRFs by relaxing the assumption that only the linear structure of the input matters: they add explicit dependencies between distant nodes in an otherwise linear-chain CRF. These dependencies are picked based on particular properties, e.g., input variables of the same value or which share other properties. In doing so, they add loops to the model's factor graph, which makes exact training and inference intractable, and leads to the use of approximation techniques such as loopy belief propagation and Gibbs sampling.

**Weight learning for finite-state transducers.**    While our approach focuses on the task of constraining the CRF distribution to a known regular language, a related task is that of learning a weighted regular language from data. This task is usually formalized as learning the weights of a weighted finite-state transducer (FST), as in e.g. Eisner (2002) with directly parameterized weights and Rastogi et al. (2016) with weights parameterized by a neural network. Despite the difference in task-setting, this task is quite similar to ours in the formal sense, and in fact our proposal can be

viewed as a particularly well-behaved special case of FST weight learning for an appropriately chosen transducer architecture and parameterization. We discuss this connection further in Section 4.3.

# 3 PRELIMINARIES AND NOTATION

As our construction involves finite-state automata and conditional random fields, we define these here and specify the notation we will use in the remainder of this work.

**Finite-state automata.** All automata are taken to be nondeterministic finite-state automata (NFAs) without epsilon transitions. Let such an NFA be defined as a 5-tuple $(\Sigma, Q, q_1, F, E)$, where $\Sigma = \{a_1, a_2, ..., a_{|\Sigma|}\}$ is a finite alphabet of symbols, $Q = \{q_1, q_2, ..., q_{|Q|}\}$ is a finite set of states, $q_1 \in Q$ is the sole starting state, $F \subseteq Q$ is a set of accepting states, and $E \subseteq Q \times \Sigma \times Q$ is a set of directed, symbol-labeled edges between states. The edges define the NFA's transition function $\Delta : Q \times \Sigma \to 2^Q$, with $r \in \Delta(q, a) \leftrightarrow (q, a, r) \in E$. An automaton is said to accept a string $s \in \Sigma^*$ iff there exists a contiguous path of edges from $q_1$ to some accepting state whose edge labels are exactly the symbols of $s$. The *language* defined by an automaton is the set of all such accepted strings. A language is *regular* if and only if it is the language of some NFA.

**Linear-chain conditional random fields.** Linear-chain conditional random fields (CRFs) Lafferty et al. (2001) are a sequence labeling model. Parameterized by $\theta$, they use a global exponential model to represent the conditional distribution over label sequences $\boldsymbol{y} = \langle y_1, y_2, ..., y_t \rangle$ conditioned on input sequences $\boldsymbol{x} = \langle x_1, x_2, ..., x_t \rangle$: $P_\theta(\boldsymbol{y} \mid \boldsymbol{x}) \propto \exp \sum_j f_\theta^j(\boldsymbol{x}, \boldsymbol{y})$, with individual observations $x_i$ coming from some observation space $X$, and outputs $y_i$ coming from some finite alphabet $Y$. In this work, we use CRFs for sequence labeling problems, but the dataset labels do not correspond directly to the CRF's outputs $y_i$. In order to avoid ambiguity, and since the term "state" already has a meaning for NFAs, we call $\boldsymbol{y}$ the CRF's *tag sequence*, and each $y_i$ a *tag*. The terms *label sequence* and *label* will thus unambiguously refer to the original dataset labels.

Each $f_\theta^j$ is a potential function of $\boldsymbol{x}$ and $\boldsymbol{y}$, parameterized by $\theta$. Importantly, in a linear-chain CRF, these potential functions are limited to two kinds: The *transition function* $g_\theta(y_i, y_{i+1})$ assigns a potential to each pair $(y_i, y_{i+1})$ of adjacent tags in $\boldsymbol{y}$, and the *emission function* $h_\theta(y_i \mid \boldsymbol{x}, i)$ assigns a potential to each possible output tag $y_i$ given the observation sequence $\boldsymbol{x}$ and its position $i$. With these, the distribution defined by a CRF is

$$P_\theta(\boldsymbol{y} \mid \boldsymbol{x}) \propto \exp \left( \sum_{i=1}^{t-1} g_\theta(y_i, y_{i+1}) + \sum_{i=1}^{t} h_\theta(\boldsymbol{x}, y_i, i) \right). \tag{1}$$

Limiting our potential functions in this way imposes a Markov assumption on our model, as potential functions can only depend on a single tag or a single pair of adjacent tags. This makes learning and inference tractable: the forward algorithm (Jurafsky & Martin, 2009) can calculate negative log-likelihood (NLL) loss during training, and the Viterbi algorithm (Viterbi, 1967; Jurafsky & Martin, 2009) can be used for inference. Both are linear in $t$, and quadratic in $|Y|$ in both time and space.

# 4 REGULAR-CONSTRAINED CRFs

Given a regular language $\mathcal{L}$, we would like to constrain a CRF to $\mathcal{L}$. We formalize this notion of constraint with conditional probabilities – a CRF constrained to $\mathcal{L}$ is described by a (further) conditionalized version of that CRF's distribution $P_\theta(\boldsymbol{y} \mid \boldsymbol{x})$, conditioned on the event that the tag sequence $\boldsymbol{y}$ is in $\mathcal{L}$. We write this distribution as

$$P_\theta(\boldsymbol{y} \mid \boldsymbol{x}, \mathcal{L}) = \begin{cases} \alpha \cdot P_\theta(\boldsymbol{y} \mid \boldsymbol{x}) & \text{if } \boldsymbol{y} \in \mathcal{L} \\ 0 & \text{otherwise,} \end{cases} \tag{2}$$

with $\alpha \geq 1$ defined as $\alpha^{-1} = \sum_{\boldsymbol{y} \in \mathcal{L}} P_\theta(\boldsymbol{y} \mid \boldsymbol{x})$.

In order to utilize this distribution for machine learning, we need to be able to compute NLL losses and perform MAP inference. As discussed in Section 3, both of these are efficiently computable for CRFs. Thus, if we can construct a separate CRF whose output distribution can be interpreted as $P(\boldsymbol{y} \mid \boldsymbol{x}, \mathcal{L})$, both of these operations will be available. We do this in the next section.

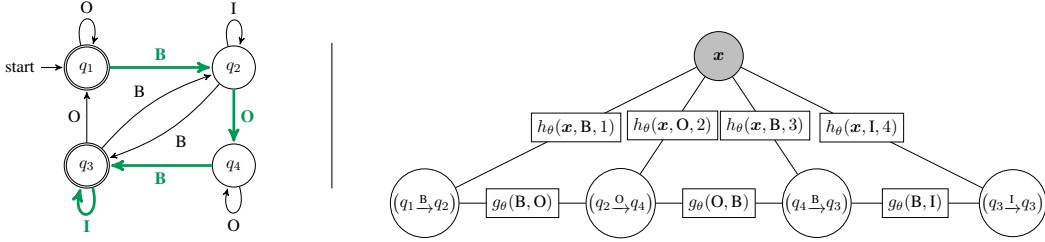

$$Y' = \left\{ \left(q_1 \xrightarrow{\text{O}} q_1\right), \left(q_1 \xrightarrow{\text{B}} q_2\right), \left(q_2 \xrightarrow{\text{I}} q_2\right), \left(q_2 \xrightarrow{\text{B}} q_3\right), \left(q_2 \xrightarrow{\text{O}} q_4\right), \left(q_3 \xrightarrow{\text{O}} q_1\right), \left(q_3 \xrightarrow{\text{B}} q_2\right), \left(q_3 \xrightarrow{\text{I}} q_3\right), \left(q_4 \xrightarrow{\text{B}} q_3\right), \left(q_4 \xrightarrow{\text{O}} q_4\right) \right\}$$

Figure 1: Example for a RegCCRF, showing NFA and unrolled factor graph. $\mathcal{L}$ describes the language (O | BI$^*$O$^*$BI$^*$)$^*$, the language of valid BIO sequences for an even number of spans. We would like to calculate $P_\theta(\boldsymbol{y} \mid \boldsymbol{x}, \mathcal{L})$ for $\boldsymbol{y} = \langle \text{B}, \text{O}, \text{B}, \text{I} \rangle$. We show an unambiguous automaton $M$ for $\mathcal{L}$ (left), and a factor graph (right) for the auxiliary CRF computing $P_\theta(\boldsymbol{y}' \mid \boldsymbol{x})$, where $\boldsymbol{y}' \in Y'^*$ corresponds to the sole accepting path of $\boldsymbol{y}$ through $M$ (marked).

## 4.1 CONSTRUCTION

Let $M := (\Sigma, Q, F, E)$ be an NFA that describes $\mathcal{L}$. We assume that $M$ is *unambiguous* – i.e., every string in $\mathcal{L}$ is accepted by exactly one path through $M$. As every NFA can be transformed into an equivalent unambiguous NFA (Mohri, 2012), this assumption involves no loss of generality. Our plan is to represent $P_\theta(\boldsymbol{y} \mid \boldsymbol{x}, \mathcal{L})$ by constructing a separate CRF with a distinct tag set, whose output sequences can be interpreted directly as paths through $M$. As $M$ is unambiguous, each label sequence in $\mathcal{L}$ corresponds to exactly one such path. We parameterize this auxiliary CRF identically to our original CRF – that is, with label-wise (not tag-wise) transition and emission functions. Thus, for all parameterizations $\theta$, both distributions $P_\theta(\boldsymbol{y} \mid \boldsymbol{x})$ and $P_\theta(\boldsymbol{y} \mid \boldsymbol{x}, \mathcal{L})$ are well defined.

There are many ways to construct such a CRF. As CRF training and inference are quadratic in the size of the tag set $Y$, we would prefer a construction which minimizes $|Y|$. However, for clarity, we will first present a conceptually simple construction, and discuss approaches to reduce $|Y|$ in Section 4.2. We start with our original CRF, parameterized by $\theta$, with tag set $Y = \Sigma$, transition function $g_\theta$, and emission function $h_\theta$, describing the probability distribution $P_\theta(\boldsymbol{y} \mid \boldsymbol{x})$, $\boldsymbol{y} \in \Sigma^*$. From this, we construct a new CRF, also parameterized by the same $\theta$, but with tag set $Y'$, transition function $g'_\theta$, and emission function $h'_\theta$. This auxiliary CRF describes the distribution $P'_\theta(\boldsymbol{y}' \mid \boldsymbol{x})$ (with $\boldsymbol{y}' \in Y'^*$), which we will be able to interpret as $P_\theta(\boldsymbol{y} \mid \boldsymbol{x}, \mathcal{L})$. The construction is as follows:

$$Y' = E \tag{3}$$

$$g'_\theta((q, a, r), (q', a', r')) = \begin{cases} g_\theta(a, a') & \text{if } r = q' \\ -\infty & \text{otherwise} \end{cases} \tag{4}$$

$$h'_\theta(\boldsymbol{x}, (q, a, r), i) = \begin{cases} -\infty & \text{if } i = 1, q \neq q_1 \\ -\infty & \text{if } i = t, r \notin F \\ h_\theta(\boldsymbol{x}, a, i) & \text{otherwise.} \end{cases} \tag{5}$$

This means that the tags of our new CRF are the edges of $M$, the transition function assigns zero probability to transitions between edges which do not pass through a shared NFA state, and the emission function assigns zero probability to tag sequences which do not begin at the starting state or end at an accepting state. Apart from these constraints, the transition and emission functions depend only on edge labels, and not on the edges themselves, and agree with the standard CRF's $g_\theta$ and $h_\theta$ when no constraints are violated.

As $M$ is unambiguous, every tag sequence $\boldsymbol{y}$ corresponds to a single path through $M$, representable as an edge sequence $\boldsymbol{\pi} = \langle \pi_1, \pi_2, ..., \pi_t \rangle, \pi_i \in E$. Since this path is a tag sequence for our auxiliary CRF, we can directly calculate the auxiliary CRF's $P'_\theta(\boldsymbol{\pi} \mid \boldsymbol{x})$. From the construction of $g'_\theta$ and $h'_\theta$, this must be equal to $P_\theta(\boldsymbol{y} \mid \boldsymbol{x}, \mathcal{L})$, as it is proportional to $P_\theta(\boldsymbol{y} \mid \boldsymbol{x})$ for $\boldsymbol{y} \in \mathcal{L}$, and zero (or, more correctly, undefined) otherwise. Figure 1 illustrates this construction with a concrete example.

## 4.2 TIME AND SPACE EFFICIENCY

As the Viterbi and forward algorithms are quadratic in $|Y|$, very large tag sets can lead to performance issues, possibly making training or inference intractible in extreme cases. Thus, we would like to characterize under which conditions a RegCCRF can be used tractibly, and identify techniques for improving performance. As $Y$ corresponds to the edges of $M$, we would like to select our unambiguous automaton $M$ to have as few edges as possible. For arbitrary languages, this problem is NP-complete (Jiang & Ravikumar, 1991), and, assuming P $\neq$ NP, is not even efficiently approximable (Gruber & Holzer, 2007). Nonetheless, for many common classes of languages, there exist approaches to obtain a tractably small automaton.

One straightforward method is to construct $M$ directly from a short unambiguous regular expression. Brüggemann-Klein & Wood (1992) present a simple algorithm for constructing an unambiguous automaton from an unambiguous regular expression, with $|Q|$ linear in the length of the expression. Using this method to construct $M$, the time- and space-complexity of Viterbi are polynomial in the length of our regular expression, with a worst-case of quartic complexity when the connectivity graph of $M$ is dense.

For many other tasks, a reasonable approach is to leverage domain knowledge about the constraints to manually construct a small unambiguous automaton. For example, if the constraints require that a particular label occurs exactly $n$ times in the output sequence, an automaton could be constructed manually to count ocurrences of that label. Multiple constraints of this type can then be composed via automaton union and intersection.

Without making changes to $M$, we can also reduce the size of $|Y|$ by adjusting our construction. Instead of making each edge of $M$ a tag, we can adopt equivalence classes of edges. Reminiscent of standard NFA minimization, we define $(q, a, r) \sim (q', a', r') \leftrightarrow (q, a) = (q', a')$. When constructing our CRF, whenever a transition would have been allowed between two edges, we allow a transition between their corresponding equivalence classes. We do the same to determine which classes are allowed as initial or final tags. As each equivalence class corresponds (non-uniquely) to a single symbol $a$, we can translate between tag sequences and strings of $\mathcal{L}$ just as before.

## 4.3 INTERPRETATION AS A WEIGHTED FINITE-STATE TRANSDUCER

While we present our model as a variation of a standard CRF which enforces regular-language constraints, an alternate characterization is as a weighted finite-state transducer with the transducer topology and weight parameterization chosen so as to yield the distribution $P_\theta(\boldsymbol{y} \mid \boldsymbol{x}, \mathcal{L})$. In order to accommodate CRF transition weights, such an approach involves weight-learning in an auxiliary automaton whose edges correspond to edge-pairs in $M$ – we give a full construction in Appendix C.

This interpretation enables direct comparison to studies on weight learning in finite-state transducers, such as Rastogi et al. (2016). While RegCCRFs can be viewed as special cases of neural-weighted FSTs, they inherit a number of useful properties from CRFs not possessed by neural-weighted automata in general. Firstly, as $|\boldsymbol{y}|$ is necessarily equal to $|\boldsymbol{x}|$, the partition function $\sum_{\boldsymbol{y} \in \mathcal{L}} P_\theta(\boldsymbol{y} \mid \boldsymbol{x}, \mathcal{L})$ is guaranteed to be finite, and $P_\theta(\boldsymbol{y} \mid \boldsymbol{x}, \mathcal{L})$ is a well-defined probability distribution for all $\theta$, which is not true for weighted transducers in general, which may admit paths with unbounded lengths and weights. Secondly, as $M$ is assumed to be unambiguous, string probabilities correspond exactly to path probabilities, allowing for exact MAP inference with the Viterbi algorithm. In contrast, finding the most probable string in the highly ambiguous automata used when learning edge weights for an unknown language is NP-Hard (Casacuberta & de la Higuera, 1999), necessitating approximation methods such as crunching (May & Knight, 2006). Finally, as each RegCCRF can be expressed as a CRF with a particular parameterization, the convexity guarantees of standard CRFs carry over, in that the loss is convex with respect to emission and transition scores. In contrast, training losses in general weighted finite-state transducers are usually nonconvex (Rastogi et al., 2016).

## 5 CONSTRAINED TRAINING VS. CONSTRAINED DECODING

Our construction suggests two possible use cases for a RegCCRF: *constrained decoding*, where a CRF is trained unconstrained, and the learned weights are then used in a RegCCRF at decoding time, and *constrained training*, where a RegCCRF is both trained and decoded with constraints. In

this section, we compare between these two approaches and a standard, *unconstrained CRF*. We assume a machine learning setting where we have access to samples from some data distribution $\widetilde{P}(\boldsymbol{x}, \boldsymbol{y})$, with each $\boldsymbol{x} \in X^*$, and each $\boldsymbol{y}$ of matching length in some regular language $\mathcal{L} \subseteq \Sigma^*$. We wish to model the conditional distribution $\widetilde{P}(\boldsymbol{y} \mid \boldsymbol{x})$ with either a CRF or a RegCCRF, by way of maximizing the model's (log) likelihood given the data distribution.

The unconstrained CRF corresponds to a CRF that has been trained, without constraints, on data from $\widetilde{P}(\boldsymbol{x}, \boldsymbol{y})$, and is used directly for inference: It makes no use of the language $\mathcal{L}$. The model's output distribution is $P_{\theta_u}(\boldsymbol{y} \mid \boldsymbol{x})$, with parameter vector $\theta_u$ minimizing the NLL objective:

$$\theta_u = \arg \min_{\theta} E_{\boldsymbol{x}, \boldsymbol{y} \sim \widetilde{P}} \left[ -\ln P_{\theta}(\boldsymbol{y} \mid \boldsymbol{x}) \right] \tag{6}$$

In constrained decoding, a CRF is trained unconstrained, but its weights are used in a RegCCRF at decoding time. The output distribution of such a model is $P_{\theta_u}(\boldsymbol{y} \mid \boldsymbol{x}, \mathcal{L})$. It is parameterized by the same parameter vector $\theta_u$ as the unconstrained CRF, as the training procedure is identical, but the output distribution is conditioned on $\boldsymbol{y} \in \mathcal{L}$.

Constrained training involves directly optimizing a RegCCRF's output distribution, avoiding any asymmetry between training and decoding time. In this case, the output distribution of the model is $P_{\theta_c}(\boldsymbol{y} \mid \boldsymbol{x}, \mathcal{L})$, where

$$\theta_c = \arg \min_{\theta} E_{\boldsymbol{x}, \boldsymbol{y} \sim \widetilde{P}} \left[ -\ln P_{\theta}(\boldsymbol{y} \mid \boldsymbol{x}, \mathcal{L}) \right] \tag{7}$$

is the parameter vector which minimizes the NLL of the RegCCRF's constrained distribution.

These three approaches form a hierarchy in terms of their ability to match the data distribution: $L_{\text{unconstrained}} \geq L_{\text{constrained decoding}} \geq L_{\text{constrained training}}$, with each $L$ corresponding to the negative log-likelihood assigned by each model to the data; see Appendix B for a proof. This suggests we should prefer the constrained training regimen.

## 6 SYNTHETIC DATA EXPERIMENTS

While constrained training cannot underperform constrained decoding, the conditions where it is strictly better depend on exactly how the transition and emission functions are parameterized, and are not easily stated in general terms. We now empirically show two simple experiments on synthetic data where the two are not equivalent.

The procedure is similar for both experiments. We specify a regular language $\mathcal{L}$, an observation alphabet $X$, and a joint data distribution $\widetilde{P}(\boldsymbol{x}, \boldsymbol{y})$ over observation sequences in $X^*$ and label sequences in $\mathcal{L}$. We then train two models, one with a RegCCRF, parameterized by $\theta_c$, and one with an unconstrained CRF, parameterized by $\theta_u$. For each model, we initialize parameters randomly, then use stochastic gradient descent to minimize NLL with $\widetilde{P}$. We directly generate samples from $\widetilde{P}$ to use as training data. After optimizing $\theta_c$ and $\theta_u$, construct a RegCCRF with $\theta_u$ for use as a constrained-decoding model, and we compare the constrained-training distribution $P_{\theta_c}(\boldsymbol{y} \mid \boldsymbol{x}, \mathcal{L})$ with the constrained-decoding distribution $P_{\theta_u}(\boldsymbol{y} \mid \boldsymbol{x}, \mathcal{L})$.

We use a simple architecture for our models, with both the transition functions $g_{\theta}$ and emission functions $h_{\theta}$ represented as parameter matrices. We list training hyperparameters in Appendix A.

### 6.1 ARBITRARILY LARGE DIFFERENCES IN LIKELIHOOD

We would like to demonstrate that, when comparing constrained training to constrained decoding in terms of likelihood, constrained training can outperform constrained decoding by an arbitrary margin. We choose $\mathcal{L} = (\texttt{ac})^* \mid (\texttt{bc})^*$ to make conditional independence particularly relevant – as every even-indexed label is $\texttt{c}$, an unconstrained CRF must model odd-indexed labels independently, while a constrained CRF can use its constraints to account for nonlocal dependencies. For simplicity, we hold the input sequence constant, with $X = \{\texttt{o}\}$.

Our approach is to construct sequences of various lengths. For $k \in \mathbb{N}$, we let our data distribution be

$$\widetilde{P}(\texttt{o}^{2k}, (\texttt{ac})^k) = \frac{3}{4} \text{ and } \widetilde{P}(\texttt{o}^{2k}, (\texttt{bc})^k) = \frac{1}{4}. \tag{8}$$

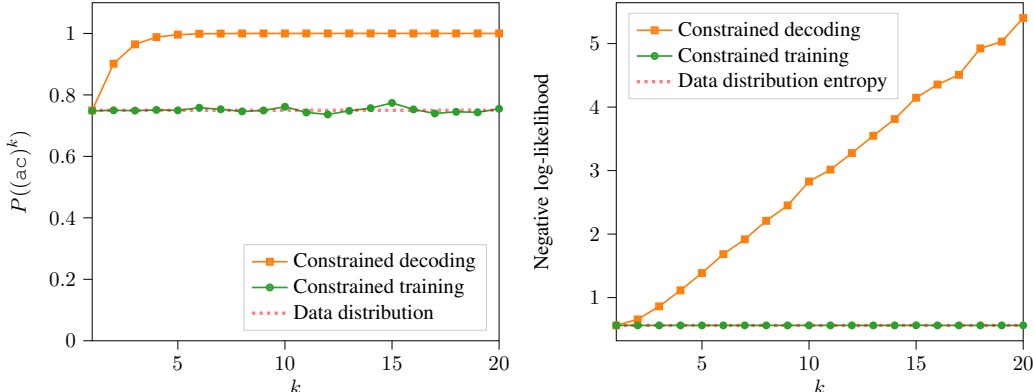

Figure 2: Model output probabilities, and NLL losses, plotted against sequence length $k$. As $k$ increases, constrained decoding becomes a progressively worse approximation for the data distribution, while constrained training is consistently able to match the data distribution.

Table 1: Output distributions for constrained decoding ($P_{\theta_u}(\boldsymbol{y} \mid \boldsymbol{x}, \mathcal{L})$) and constrained training ($P_{\theta_c}(\boldsymbol{y} \mid \boldsymbol{x}, \mathcal{L})$), compared to the data distribution $\widetilde{P}(\boldsymbol{y} \mid \boldsymbol{x})$. Constrained decoding cannot learn the data distribution exactly, and yields a mode which disagrees with that of the data distribution.

| $\boldsymbol{y}$ | $\widetilde{P}(\boldsymbol{y} \mid \boldsymbol{x})$ | $P_{\theta_u}(\boldsymbol{y} \mid \boldsymbol{x}, \mathcal{L})$ | $P_{\theta_c}(\boldsymbol{y} \mid \boldsymbol{x}, \mathcal{L})$ |
|---|---|---|---|
| acd | **0.4** | 0.32 | **0.40** |
| bcd | 0.3 | **0.48** | 0.30 |
| bce | 0.3 | 0.20 | 0.30 |

As the marginal distributions for odd-indexed characters are not independent, an unconstrained CRF cannot exactly represent the distribution $\widetilde{P}$. We train and evaluate individual models for each sequence length $k$. Figure 2 plots model probabilities and NLL losses for various $k$. We see that, regardless of $k$, $P_{\theta_c}(\boldsymbol{y} \mid \boldsymbol{x}, \mathcal{L})$ is able to match $\widetilde{P}(\boldsymbol{y} \mid \boldsymbol{x})$ almost exactly, with only small deviations due to sampling noise in SGD. On the other hand, as sequence length increases, $P_{\theta_u}(\boldsymbol{y} \mid \boldsymbol{x}, \mathcal{L})$ becomes progressively "lop-sided", assigning almost all of its probability mass to the string $(\mathtt{ac})^k$. This behavior is reflected in the models' likelihoods – constrained training stays at near-constant likelihood for all $k$, while the negative log-likelihood of constrained decoding grows linearly with $k$.

## 6.2 DIFFERENCES IN MAP INFERENCE

We show here that constrained training and constrained decoding can disagree about which label sequence they deem most likely. Furthermore, in this case, MAP inference agrees with the data distribution's mode for constrained training, but not for constrained decoding. To do this, we construct a fixed-length output language $\mathcal{L} = \mathtt{acd} \mid \mathtt{bcd} \mid \mathtt{bce}$, where an unconstrained CRF is limited by the Markov property to predict $\boldsymbol{y}$'s prefix and suffix independently, and choose a data distribution which violates this independence assumption. We select our data distribution,

$$\widetilde{P}(\mathtt{ooo}, \mathtt{acd}) = 0.4 \text{ and } \widetilde{P}(\mathtt{ooo}, \mathtt{bcd}) = 0.3 \text{ and } \widetilde{P}(\mathtt{ooo}, \mathtt{bce}) = 0.3, \quad (9)$$

to be close to uniform, but with one label sequence holding the slight majority, and we ensure that the majority label sequence is *not* the label sequence with both the majority prefix and the majority suffix (i.e. $\mathtt{bcd}$). As before, we hold the observation sequence as a constant ($\mathtt{ooo}$). We train a constrained and an unconstrained CRF to convergence, and compare $P_{\theta_u}(\boldsymbol{y} \mid \boldsymbol{x}, \mathcal{L})$ to $P_{\theta_c}(\boldsymbol{y} \mid \boldsymbol{x}, \mathcal{L})$.

Table 1 shows $P_{\theta_u}(\boldsymbol{y} \mid \boldsymbol{x}, \mathcal{L})$ and $P_{\theta_c}(\boldsymbol{y} \mid \boldsymbol{x}, \mathcal{L})$ as they compare to $\widetilde{P}(\boldsymbol{y} \mid \boldsymbol{x})$. We find that, while the mode of $\widetilde{P}(\boldsymbol{y} \mid \boldsymbol{x})$ is $\mathtt{acd}$, with probability of 0.4, the mode of constrained decoding distribution $P_{\theta_u}(\boldsymbol{y} \mid \boldsymbol{x}, \mathcal{L})$ is $\mathtt{bcd}$, the string with the majority prefix and the majority suffix, to which the model assigns a probability of 0.48. Conversely, the constrained training distribution $P_{\theta_c}(\boldsymbol{y} \mid \boldsymbol{x}, \mathcal{L})$ matches the data distribution almost exactly, and predicts the correct mode.

## 7    REAL-WORLD DATA EXPERIMENT: SEMANTIC ROLE LABELING

**Task.**    As a final experiment, we apply our RegCCRF to the NLP task of semantic role labeling (SRL) in the popular PropBank framework (Palmer et al., 2005). In line with previous work, we adopt the *known-predicate setting*, where events are given and the task is to mark token spans as (types of) event participants. PropBank assumes 7 semantic *core roles* (ARG0 through ARG5 plus ARGA) plus 21 *non-core roles* for modifiers such as times or locations. For example, in [$_{ARG0}$ *Peter*] **saw** [$_{ARG1}$ *Paul*] [$_{ARGM-TMP}$ *yesterday*], the argument labels inform us who does the seeing (ARG0), who is seen (ARG1), and when the event took place (ARGM-TMP). In addition, role spans may be labeled as *continuations* of previous role spans, or as *references* to another role span in the sentence. SRL can be framed naturally as a sequence labeling task (He et al., 2017). However, the task comes with a number of hard constraints that are not automatically satisfied by standard CRFs, namely: (1) Each core role may occur at most once per event; (2) for continuations, the span type must occur previously in the sentence; (3) for references, the span type must occur elsewhere in the sentence.

**Data.**    In line with previous work (Ouchi et al., 2018), we work with the OntoNotes corpus as used in the CoNLL 2012 shared task[1] (Weischedel et al., 2011; Pradhan et al., 2012), whose training set comprises 66 roles (7 core roles, 21 non-core roles, 19 continuation types, and 19 reference types).

**RegCCRF Models.**    To encode the three constraints listed above in a RegCCRF, we define a regular language describing valid BIO sequences (Ramshaw & Marcus, 1999) over the 66 roles. A minimal unambiguous NFA for this language has more than $2^2 \cdot 3^{19}$ states, which is too large to run the Viterbi algorithm on our hardware. However, as many labels are very rare, we can shrink our automaton by discarding them at the cost of imperfect recall. We achieve further reductions in size by ignoring constraints on reference roles, treating them identically to non-core roles. Our final automaton recognizes 5 core role types (ARG0 through ARG4), 17 non-core / reference roles, and one continuation role type (for ARG1). This automaton has 672 states, yielding a RegCCRF with 2592 tags. A description of our procedure for constructing this automaton can be found in Appendix D.

Our model architecture is given by this RegCCRF, with emission scores provided by a linear projection of the output of a pretrained RoBERTa network Liu et al. (2019). In order to provide the model with event information, the given predicates are prefixed by a special reserved token in the input sequence. RoBERTa parameters are fine-tuned during the learning of transition scores and projection weights. We perform experiments with both constrained training and constrained decoding settings – we will refer to these as *ConstrTr* and *ConstrDec* respectively. A full description of the training procedure, including training times, is provided in Appendix A. As RegCCRF loss is only finite for label sequences in $\mathcal{L}$, we must ensure that our training data do not violate our constraints. We discard some roles, as described above, by simply removing the offending labels from the training data. In six cases, training instances directly conflict with the constraints specified — all cases involve continuation roles missing a valid preceding role. We discard these instances for *ConstrTr*.

**CRF Baselines.**    As baseline models, we use the same architecture, but with a standard CRF replacing the RegCCRF. Since we are not limited by GPU memory for CRFs, we are optionally able to include all role types present in the training set, using the complete training set. We present two CRF baseline models: *CRF-full*, which is trained on all role-types from the training set, and *CRF-reduced*, which includes the same subset of roles as the RegCCRF models. For *CRF-reduced*, we use the same learned weights as for CD, but we decode without constraints.

**Results and analysis.**    We evaluate our models on the evaluation partition, and measure performance using $F_1$ score for exact span matches. For comparability with prior work, we use the evaluation script[2] for the CoNLL-2005 shared task (Carreras & Màrquez, 2005). These results, averaged over five trials, are presented in Table 2. Excepting *CRF-reduced*, all of our models outperform the existing state-of-the-art ensemble model Ouchi et al. (2018). We ascribe this improvement over the existing literature to our use of RoBERTa – prior work in SRL relies on ELMo (Peters et al., 2018), which tends to underperform transformer-based models on downstream tasks (Devlin et al., 2019).

---

[1]As downloaded from `https://catalog.ldc.upenn.edu/LDC2013T19`, and preprocessed according to `https://cemantix.org/data/ontonotes.html`

[2]As available from `https://www.cs.upc.edu/~srlconll/soft.html`.

Table 2: Results from our experiments (averaged over 5 trials), along with selected reported results from recent literature. We rank of our models by precision, recall, and $F_1$ score – rankings differ if and only if the difference is significant at $p < 0.05$ (two-tailed), as measured by a permutation test.

|  | Model | Precision[rank] | Recall[rank] | $F_1$[rank] |
|---|---|---|---|---|
| Our experiments | RoBERTa + CRF (*CRF-full*) | $86.82^{(2)}$ | $87.73^{(1)}$ | $87.27^{(2)}$ |
|  | RoBERTa + CRF (*CRF-reduced*) | $87.33^{(1)}$ | $85.95^{(3)}$ | $86.63^{(3)}$ |
|  | RoBERTa + RegCCRF (*ConstrDec*) | $87.28^{(1)}$ | $87.13^{(2)}$ | $87.20^{(2)}$ |
|  | RoBERTa + RegCCRF (*ConstrTr*) | $87.22^{(1)}$ | $\mathbf{87.79^{(1)}}$ | $\mathbf{87.51^{(1)}}$ |
| Results from literature | He et al. (2017) | — | — | 85.5 |
|  | Ouchi et al. (2018) | 87.1 | 85.3 | 86.2 |
|  | Ouchi et al. (2018) (ensemble) | **88.5** | 85.5 | 87.0 |
|  | Li et al. (2019) | 85.7 | 86.3 | 86.0 |

Of our models, *ConstrTr* significantly[3] outperforms the others in $F_1$ score and yields a new SOTA for SRL on OntoNotes, in line with expectations from theoretical analysis and on synthetic data. Our other three models show trade-offs between precision and recall, wherein *CRF-full* outperforms the other two in recall and underperforms them in precision. This is not surprising, as *CRF-full* is the only model capable of predicting rare role types. The other models, which use the reduced tag sets, have a theoretical maximum of 99% recall. Interestingly, when comparing between the three models that use the reduced tag sets, we find a statistically significant interaction between the constraint setting and model recall, but not between constraints and model precision: *ConstrTr* has significantly higher recall than *ConstrDec*, which in turn significantly beats *CRF-reduced* in recall, but there are no statistically significant differences between these models' precisions.

For our unconstrained models, *CRF-full* and *CRF-reduced*, the constraints specified in our automaton are violated in 0.81% and 0.84% of all output sequences respectively.[4] While this number is small, it should *not* be interpreted to mean that constraints are useless for almost all instances – as shown in Section 6.2, constraints during training can affect MAP inference even when none of the alternatives violate constraints.

## 8 CONCLUSION AND FUTURE WORK

We have presented a method to constrain the output of CRFs to a regular language. Our construction permits constraints to be used at training or prediction time; both theoretically and empirically, training-time constraining better captures the data distribution. Conceptually, our approach constitutes a novel bridge between constrained CRFs and neural-weighted FSTs.

Future work could target enhancing the model's expressibility, either by allowing constraints to depend explicitly on the input as regular relations, or by investigating non-binary constraints, i.e., regular language-based constraints with learnable weights. Additionally, regular language induction (e.g. Dunay et al. (1994); Bartoli et al. (2016)) could be used to learn languages automatically, reducing manual specification and identifying non-obvious constraints. Another avenue for continuing research lies in identifying further applications for RegCCRFs. The NLP task of relation extraction could be a fruitful target – RegCCRFs offer a mechanism to make the proposal of a relation conditional on the presence of the right number and type of arguments. While our construction cannot be lifted directly to context-free languages due to the unbounded state space of the corresponding pushdown automata, context-free language can be approximated by regular languages (Mohri & Nederhof, 2001). On this basis, for example, a RegCCRF backed by a regular language describing trees of a limited depth could also be applied to tasks with context-free constraints.

To encourage the use of RegCCRFs, we provide an implementation as a Python library under the Apache 2.0 license which can be used as a drop-in replacement for standard CRFs in PyTorch.[5]

---

[3]All significance results are at the $p < 0.05$ level (two-tailed), as measured by a permutation test over the five trials of each model.

[4]For *CRF-full*, we only count violations of constraints for those roles that our automaton accounts for.

[5]Available at www.ims.uni-stuttgart.de/en/research/resources/tools/regccrf/

## ACKNOWLEDGMENTS

This work is supported by IBM Research AI through the IBM AI Horizons Network.

## REPRODUCIBILITY STATEMENT

To ensure reproducibility, we have released all code for the RegCCRF model as an open-source Python 3 library under the Apache 2.0 license, which is included in the supplementary materials. Additionally, we include Python scripts for reproducing all experiments presented in the paper, detailed descriptions of our datasets and preprocessing steps, and training logs. Furthermore, Appendix A lists all model hyperparameters, details the preprocessing steps taken for our experiments, and specifies the hardware used for our experiments along with average training and inference times for each experiment.

## ETHICS STATEMENT

The research in this paper is fundamental in the sense that it enables machine learning models to better represent data and limit the search space at inference and learning time. It therefore does not in and of itself represent additional ethical risks on top of the previous work we build upon.

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

## A    EXPERIMENTAL DESIGN

This appendix details the training procedures and hyperparameter choices for our experiments. These are summarized in Table 3. Full code for all experiments, along with training logs, are also included in the supplementary materials.

### A.1    CRFs

For all CRFs and RegCCRFs, transition potentials were initialized randomly from a normal distribution with mean zero and standard deviation 0.1. No CRFs or RegCCRFs employed special start- or end-transitions – that is, we did not insert any additional beginning-of-sequence or end-of-sequence tags for the Viterbi or forward algorithms.

Table 3: Summary of hyperparameters for our models and experiments.

| CRFs | Transition score initialization | $\mathcal{N}(0, 0.1)$ |
|---|---|---|
| **Synthetic data experiments** | Emission score initialization | PyTorch default |
| | Optimizer | SGD |
| | Training iterations | 5000 |
| | Batch size | 50 |
| | Initial learning rate | 1.0 |
| | Learning rate decay | 10% every 100 steps |
| **SRL experiments** | RoBERTa weights | `roberta-base` |
| | Projection weight and bias initialization | PyTorch default |
| | Optimizer | Adam |
| | Learning rate | $10^{-5}$ |
| | Batch size | 2 |
| | Gradient accumulation | 4 batches |

## A.2 SYNTHETIC DATA EXPERIMENTS – TRAINING PROCEDURE

For both synthetic data experiments, the emission potentials were represented explicitly for each position as trainable parameters – since the observation sequence was constant in all experiments, these did not depend on $x$.

Parameters were initialized randomly using PyTorch default initialization, and optimized using stochastic gradient descent. To ensure fast convergence to a stable distribution, we employed learning rate decay – learning rate was initially set to 1.0, and reduced by 10% every 100 training steps.

We trained all models for a total of 5000 steps with a batch size of 50. All models were trained on CPUs. For the experiment described in Section 6.1, we trained separate models for each $k$ – the total training time for this experiment was approximately 35 minutes. The experiment described in Section 6.2 completed training in approximately 30 seconds.

## A.3 SEMANTIC ROLE LABELING – TRAINING PROCEDURE

In the semantic role labeling (SRL) experiments, we incorporated a pretrained RoBERTa network (Liu et al., 2019) – the implementation and weights for this model were obtained using the `roberta-base` model from the Hugging Face transformers library (Wolf et al., 2020). RoBERTa embeddings were projected down to transmission scores using a linear layer with a bias – projection weights and biases were initialized using the PyTorch default initialization.

Input tokens were sub-tokenized using RoBERTa's tokenizer. The marked predicate in each sentence was prefixed by a special `<unk>` token. During training, for efficiency reasons, we excluded all sentences with 120 or more subtokens – this amounted to 0.23% of all training instances. We nonetheless predicted on all instances, regardless of length.

We optimized models using the Adam optimizer (Kingma & Ba, 2015) with a learning rate of $10^{-5}$. We fine-tune RoBERTa parameters and learn the projection and RegCCRF weights jointly. For performance reasons, batch size was set to 2, but we utilized gradient accumulation over groups of 4 batches to simulate a batch size of 8.

We utilized early stopping to avoid overfitting. Every 5000 training steps, we approximated our model's $F_1$ score against a subset of the provided development partition, using a simplified reimplementation of the official evaluation script. Each time we exceeded the previous best $F_1$ score for a model, we saved all model weights to disk. After 50 such evaluations with no improvement, we terminated training, and used the last saved copy of model weights for final evaluation.

We performed all SRL experiments on GeForce GTX 1080 Ti GPUs. Each experiment used a single GPU. Training took an average of 88 hours for RegCCRF models with constrained training, 23 hours for RegCCRF with constrained decoding, and 24 hours for CRF baseline models. Inference on the complete test set took an average of 18 minutes 55 seconds for CT and CD, and an average

of 55 seconds for CRF-full and CRF-reduced. All training logs with timestamps are included in the supplementary materials.

## B  PROOF OF CONSTRAINED TRAINING INEQUALITY

In this appendix, we prove the inequality presented in section 5: when compared by NLL against the data distribution, $L_{\text{unconstrained}} \geq L_{\text{constrained decoding}} \geq L_{\text{constrained training}}$, with each $L$ corresponding to that model's negative log-likelihood. We first prove the left side of this inequality, comparing an unconstrained CRF to constrained decoding, and then prove the right side, comparing constrained decoding to constrained training. We use the notation introduced in Section 5.

**Theorem 1.** *For arbitrary $\theta$:* $E_{\boldsymbol{x},\boldsymbol{y} \sim \widetilde{P}} \left[ -\ln P_\theta(\boldsymbol{y} \mid \boldsymbol{x}) \right] \geq E_{\boldsymbol{x},\boldsymbol{y} \sim \widetilde{P}} \left[ -\ln P_\theta(\boldsymbol{y} \mid \boldsymbol{x}, \mathcal{L}) \right]$

Here we compare the distributions $P_\theta(\boldsymbol{y} \mid \boldsymbol{x})$ and $P_\theta(\boldsymbol{y} \mid \boldsymbol{x}, \mathcal{L})$. We wish to demonstrate that $P_\theta(\boldsymbol{y} \mid \boldsymbol{x})$ can never achieve lower NLL than $P_\theta(\boldsymbol{y} \mid \boldsymbol{x}, \mathcal{L})$, and that the two distributions achieve identical NLL only when $P_\theta(\boldsymbol{y} \mid \boldsymbol{x}) = P_\theta(\boldsymbol{y} \mid \boldsymbol{x}, \mathcal{L})$ i.e. when constraints have no effect. Of note, this proof is valid for *all* parameterizations $\theta$, and not just for $\theta_u$.

*Proof.* Since every $\boldsymbol{y}$ in $\widetilde{P}$ is in $\mathcal{L}$,

$$P_\theta(\boldsymbol{y} \mid \boldsymbol{x}, \mathcal{L}) = \alpha \cdot P_\theta(\boldsymbol{y} \mid \boldsymbol{x}), \tag{10}$$

with $\alpha \geq 1$. Thus, the NLL of the regular-constrained CRF is

$$E_{\boldsymbol{x},\boldsymbol{y} \sim \widetilde{P}} \left[ -\ln P_\theta(\boldsymbol{y} \mid \boldsymbol{x}, \mathcal{L}) \right] = E_{\boldsymbol{x},\boldsymbol{y} \sim \widetilde{P}} \left[ -\ln P_\theta(\boldsymbol{y} \mid \boldsymbol{x}) \right] - \ln \alpha. \tag{11}$$

This differs from the NLL of the unconstrained CRF only by the term $-\ln \alpha$. As $\alpha \geq 1$, the regular-constrained CRF's NLL is less than or equal to that of the unconstrained CRF, with equality only when $\alpha = 1$ and therefore $P_\theta(\boldsymbol{y} \mid \boldsymbol{x}) = P_\theta(\boldsymbol{y} \mid \boldsymbol{x}, \mathcal{L})$.  ∎

**Theorem 2.** $E_{\boldsymbol{x},\boldsymbol{y} \sim \widetilde{P}} \left[ -\ln P_{\theta_u}(\boldsymbol{y} \mid \boldsymbol{x}, \mathcal{L}) \right] \geq E_{\boldsymbol{x},\boldsymbol{y} \sim \widetilde{P}} \left[ -\ln P_{\theta_c}(\boldsymbol{y} \mid \boldsymbol{x}, \mathcal{L}) \right]$

In this case, we compare the distributions $P_{\theta_u}(\boldsymbol{y} \mid \boldsymbol{x}, \mathcal{L})$ and $P_{\theta_c}(\boldsymbol{y} \mid \boldsymbol{x}, \mathcal{L})$. We will demonstrate that the former cannot achieve a lower NLL against the data distribution than the latter.

*Proof.* This follows directly from our definitions, as we define $\theta_c$ to minimize the NLL of $P_\theta(\boldsymbol{y} \mid \boldsymbol{x}, \mathcal{L})$ against the data distribution. Thus, $P_{\theta_u}(\boldsymbol{y} \mid \boldsymbol{x}, \mathcal{L})$ could never yield a lower NLL than $P_{\theta_c}(\boldsymbol{y} \mid \boldsymbol{x}, \mathcal{L})$, as that would contradict our definition of $\theta_c$.  ∎

## C  CONSTRUCTION AS WEIGHTED FST

In this appendix, we present a construction of the RegCCRF as a weighted finite-state transducer with weight sharing. We do this by first specifying the transducer topology used, and then specifying how edge weights are parameterized in terms of $\theta$. The resulting transducer yields an identical distribution to that of the CRF-based construction, $P_\theta(\boldsymbol{y} \mid \boldsymbol{x}, \mathcal{L})$.

### C.1  TRANSDUCER TOPOLOGY

Starting from $\mathcal{L}$, we define $\ddot{\mathcal{L}}$ to be the regular language of bigram sequences for the strings in $\mathcal{L}$, i.e.,

$$\ddot{\mathcal{L}} = \left\{ \left\langle (s_1, s_2), (s_2, s_3), ..., (s_{|\boldsymbol{s}|-1}, s_{|\boldsymbol{s}|}), (s_{|\boldsymbol{s}|}, \$) \right\rangle \mid \boldsymbol{s} \in \mathcal{L} \right\}, \tag{12}$$

with $\$$ acting as a end-of-string symbol. We let $\ddot{M}$ be an unambiguous FSA for the language $\ddot{\mathcal{L}}$, and choose to interpret this automaton as a finite-state transducer by stipulating that all edges should accept any symbol in the input language (but only one symbol per transition, and without allowing epsilon transitions). This unweighted transducer will be used as the topology for our weighted finite-state transducer.

## C.2 Edge weights

In line with Rastogi et al. (2016), we would like to assign weights to the edges of our transducer $\ddot{M}$ with a neural network. In order to obtain the same distribution as from our CRF-based construction, these weights must be parameterized in terms of our transition function $g_\theta$ and emission function $h_\theta$. For each edge in $\ddot{M}$, the weight depends only on the emitted bigram, the input sequence, and the index of the current input symbol – the weight does not depend on the FST states. For a symbol bigram $(a, b)$, input sequence $\boldsymbol{x}$, and index $i$, the edge weight is equal to

$$W_{a,b} = \begin{cases} g_\theta(a, b) + h_\theta(\boldsymbol{x}, a, i) & b \neq \$ \\ h_\theta(\boldsymbol{x}, a, i) & \text{otherwise} \end{cases}. \tag{13}$$

Each string in $\mathcal{L}$ corresponds bijectively to exactly one bigram sequence in $\ddot{\mathcal{L}}$, which corresponds bijectively to exactly one accepting path in $\ddot{M}$ – this path's weight (in the Log semiring) is equal to the unscaled probability produced by our CRF construction, and so the weighted FST, interpreted as a probability distribution, yields the distribution $P_\theta(\boldsymbol{y} \mid \boldsymbol{x}, \mathcal{L})$.

## D Automaton construction for semantic role labeling

In this appendix, we describe how we generate the automaton architecture for our semantic role labeling experiments. While our experiments used 5 core-roles, 17 non-core roles, and one continuation role, we discuss here a generalized setting with arbitrary sets of core, noncore, and continuations of core roles.

Algorithm 1 provides pseudocode for our construction. The core idea is to use subsets of core roles as NFA states, so that we can keep track of which core roles have already occurred. Additional states are used in order to ensure all strings are valid BIO sequences.

**Data:** Sets $\mathcal{R}_{\text{core}}$, $\mathcal{R}_{\text{noncore}}$, and $\mathcal{R}_{\text{continuation}}$, of core, noncore, and continuation roles, respectively

**Result:** A finite-state automaton $M = (\Sigma, Q, q_1, F, E)$, parameterized as described in Section 3

$\Sigma \leftarrow \{\textsc{Outside}\} \cup (\{\textsc{Begin}, \textsc{Inside}\} \times (\mathcal{R}_{\text{core}} \cup \mathcal{R}_{\text{noncore}} \cup \mathcal{R}_{\text{continuation}}))$;

$Q \leftarrow \varnothing$;

$q_1 \leftarrow \varnothing$;

$F \leftarrow \varnothing$;

$E \leftarrow \varnothing$;

**for** $p \in 2^{\mathcal{R}_{core}}$ **do**

    $Q \leftarrow Q \cup \{p\}$;

    $F \leftarrow F \cup \{p\}$;

    $E \leftarrow E \cup \{(p, \textsc{Outside}, p)\}$;

    **for** $r \in \mathcal{R}_{noncore}$ **do**

        $s \leftarrow (r, p)$;

        $Q \leftarrow Q \cup \{s\}$;

        $E \leftarrow E \cup \{(p, (\textsc{Begin}, r), p), (p, (\textsc{Begin}, r), s)\}$;

        $E \leftarrow E \cup \{(r, (\textsc{Inside}, r), s), (r, (\textsc{Inside}, r), p)\}$;

    **end**

    **for** $r \in \mathcal{R}_{continuation}$ **do**

        **if** *The core role corresponding to $r$ is in $p$* **then**

            $s \leftarrow (r, p)$;

            $Q \leftarrow Q \cup \{s\}$;

            $E \leftarrow E \cup \{(p, (\textsc{Begin}, r), p), (p, (\textsc{Begin}, r), s)\}$;

            $E \leftarrow E \cup \{(r, (\textsc{Inside}, r), s), (r, (\textsc{Inside}, r), p)\}$;

        **end**

    **end**

    **for** $r \in (\mathcal{R}_{core} \setminus p)$ **do**

        $s \leftarrow (r, p)$;

        $t \leftarrow p \cup \{r\}$;

        $Q \leftarrow Q \cup \{s\}$;

        $E \leftarrow E \cup \{(p, (\textsc{Begin}, r), s), (p, (\textsc{Begin}, r), t)\}$;

        $E \leftarrow E \cup \{(s, (\textsc{Inside}, r), s), (s, (\textsc{Inside}, r), t)\}$;

    **end**

**end**

**return** $(\Sigma, Q, q_1, F, E)$

**Algorithm 1:** Construction of an FSA from given sets of core, noncore, and continuation roles. To represent BIO labels, we use tuples of the form $(\textsc{Begin}, \texttt{<roleType>})$ for B labels, tuples of the form $(\textsc{Inside}, \texttt{<roleType>})$ for I labels, and the symbol $\textsc{Outside}$ for the sole O label.

