# OpenReview forum: "Constraining Linear-chain CRFs to Regular Languages"
_ICLR.cc/2022/Conference — ICLR 2022 Poster_

### Official Review · Reviewer_qQNs · 2021-10-29

**Correctness:** 2
**Technical Novelty And Significance:** 2
**Empirical Novelty And Significance:** 2
**Recommendation:** 5
**Confidence:** 4

**Main Review:**

Strength:

1. Consider the problem of external constraints in the CRF model and propose a solution to address the nonlocal dependency issue.

2. Give a detailed mathematical description and proof of the proposed method.

Weakness:

1. The author mentioned that RegCCRF can incorporate their constraints during training, while related models only enforce constraints during decoding, but no matter in theory, I can't see what gains can be brought to the model by using constraints during training (in the current version). Because maximum likelihood estimation is used on the golden data during training, the labels will strictly follow the constraints, the constraints cannot bring any redundant information, so I don't see any advantages for supervised training under general data situations. On the contrary, I think it may be useful for training in unsupervised situations, but it has not been studied.

2. The article claims that constrained training is substantially better than constrained decoding in practice, but unfortunately from the actual results in Table 2, it is obviously not in line with this point. The difference between the two is 0.31. Because the SRL test set is not large, this result is far from significant.

3. For RegCCRF, what about the training efficiency and decoding efficiency, and how does it perform compared to traditional CRF? None of these questions have been answered in this article, so this obviously cannot make RegCCRF a replacement for traditional CRF.

4. The paper mentioned a lot of related work, but no comparisons were made in the experiment, such as Semi-Markov CRF, Skip-chain CRF, and constrained decoding (constrained beam search), which makes the performance of RegCCRF difficult to discuss.

5. Since the added tag-wise CRF can be viewed as a particularly well-behaved special case of FST weight learning for an appropriately chosen transducer architecture and parameterization, a baseline that needs to be compared is to directly use an RNN network to simulate FST, its speed is obviously faster and can be trained.

6. The results reported by the article on the SRL are only an improvement from the baseline, which is not significant compared to the traditional CRF model, and the results have not reached SOTA.


=======
I have read the response and it has addressed parts of my concerns, so I raised my score.


**Summary Of The Paper:**

This paper claims to propose a generalization version of CRF, regular-constrained CRF (RegCCRF). Compared with traditional CRF, it can not only model local interdependencies but also incorporate non-local constraints for the model. Specifically, by specifying the space of possible output structures as a regular language, assigns zero probability to all label sequences not in language to achieve the goal.
The paper spent a lot of space analyzing the difference between the proposed RegCCRF and constrained decoding, Markov relaxations methods, and finite-state transducers, and gave two settings: constrained training and constrained decoding. But as far as the actual implementation is concerned, only the introduction of a new tag-wise CRF, combined with the traditional label-wise CRF, is not much novel. In addition, the experimental part is seriously lacking, and there is no adequate experimental comparison and ablation study of the proposed method.

**Summary Of The Review:**

Although the discussion is very interesting, the current version of the experiment did not meet the requirements for publication, and many parts of the method were not properly studied.

---

> ### Author Response · Authors · 2021-11-11
> **Response to Reviewer qQNs (part 1)**
>
> Thank you for your comments. We disagree with your assessment, and
> feel you may have missed some points of our paper.  We hope
> that we can clarify these points.
>
> >The author mentioned that RegCCRF can incorporate their constraints during training, while related models only enforce constraints during decoding, but no matter in theory, I can't see what gains can be brought to the model by using constraints during training (in the current version). Because maximum likelihood estimation is used on the golden data during training, the labels will strictly follow the constraints, the constraints cannot bring any redundant information, so I don't see any advantages for supervised training under general data situations. On the contrary, I think it may be useful for training in unsupervised situations, but it has not been studied.
>
> We have demonstrated the advantages of training with constraints quite thoroughly, with a mathematical
> proof, demonstrations with synthetic data, and empirical results
> with significance tests. One way of conceptualizing this phenomenon
> is as follows: when a CRF is trained unconstrained on data which
> follows nonlocal constraints, the CRF will nonetheless use some
> model capacity to try to approximate these constraints in its output
> distribution, and this leaves less model capacity to learn other
> statistical properties of the data. When the model is trained with
> constraints, the CRF does not need to use any model capacity to
> account for these constraints, and can therefore utilize the
> entirety of its capacity for capturing other statistical properties.
>
> >The article claims that constrained training is substantially better than constrained decoding in practice, but unfortunately from the actual results in Table 2, it is obviously not in line with this point. The difference between the two is 0.31. Because the SRL test set is not large, this result is far from significant.
>
> While the numerical differences between our models' F1 scores are modest, the
> improvements seen in the constrained training regime versus the other
> models are nonetheless statistically significant. As we note on page
> 9, we train and evaluate each architecture five times, and use a
> two-tailed permutation test to measure the statistical significance
> of our model comparisons. We find that the constrained training
> regime outperforms the plain CRF at a significance level of
> $p=0.0186$ and that it outperforms the constrained decoding regime
> at a significance level of $p=0.0038$. We will update the submission
> to report these significance tests more prominently.
>
> >For RegCCRF, what about the training efficiency and decoding efficiency, and how does it perform compared to traditional CRF? None of these questions have been answered in this article, so this obviously cannot make RegCCRF a replacement for traditional CRF.
>
> You raise a good point in that these questions could be more thoroughly addressed. We do report average training time for all
> models for the SRL task in Appendix A.3: The CRF took an average of
> 24 hours to train, the RegCCRF with constrained decoding took an
> average of 23 hours to train, and the RegCCRF with constrained
> training took an average of 88 hours, all on identical hardware.
> We do not currently report inference time, but we will do so in the
> final version of this paper. The constrained training and
> constrained decoding settings have identical inference procedures,
> and take an average of 18 minutes and 55 seconds to perform inference
> for the Ontonotes 2012 evaluation partition.  The CRF model is
> substantially faster, taking an average of 55 seconds. We will
> update our submission to include a more detailed discussion of these performance
> tradeoffs in the main body.
>
> >The paper mentioned a lot of related work, but no comparisons were made in the experiment, such as Semi-Markov CRF, Skip-chain CRF, and constrained decoding (constrained beam search), which makes the performance of RegCCRF difficult to discuss.
>
> We actually do make direct comparisons to related work incorporating constrained decoding, --
> Ouchi et al. (2018) use greedy search to incorporate constraints
> during decoding, Li et al. (2019) use a dynamic-programming-based
> decoder to enforce constraints at decoding time, and He et al.
> (2017) used constrained $A^*$ decoding. None of the results we compare
> to use beam search during decoding, as the ubiquity of CRF-based
> models for this task makes $A^*$-style exact search a more appealing alternative. As we discuss in the
> related work section, skip-chain CRFs and semi-Markov CRFs are more
> suitable in settings where long-distance dependencies are not known
> a priori and must be learned by models, while RegCCRFs are
> applicable in settings with known constraints -- this makes a direct
> comparison of these models difficult.

---

> > ### Author Response · Authors · 2021-11-11
> > **Response to Reviewer qQNs (part 2)**
> >
> > >Since the added tag-wise CRF can be viewed as a particularly well-behaved special case of FST weight learning for an appropriately chosen transducer architecture and parameterization, a baseline that needs to be compared is to directly use an RNN network to simulate FST, its speed is obviously faster and can be trained.
> >
> > Could you please clarify this point, particularly with regards to simulating an FST with an RNN? If you mean to suggest that we should have compared against a more typical neural-weighted FST, we address this in our response to reviewer 82Jb -- we did not identify a particular model setup that would make a reasonable baseline.
> >
> > >The results reported by the article on the SRL are only an improvement from the baseline, which is not significant compared to the traditional CRF model, and the results have not reached SOTA.
> >
> > At the time of submission, our results exceeded the best claimed results on the Ontonotes
> > 2012 SRL task.  After the submission deadline for this conference, a preprint (Zhang et al. 2021)
> > was posted to arXiv.org which claims better results on the same dataset -- should this preprint be accepted
> > at a peer-reviewed venue before the camera-ready deadline, we will update our wording accordingly.
> > Regardless, we don't see SOTA results on Ontonotes 2012 to be a central contribution of this paper,
> > so much as part of our empirical validation of our theoretical contributions.
> > As we have already discussed, our model's improvement over a CRF baseline
> > is in fact statistically significant at a $p<.05$ level.

---

### Official Review · Reviewer_ephA · 2021-11-01

**Correctness:** 4
**Technical Novelty And Significance:** 3
**Empirical Novelty And Significance:** 3
**Recommendation:** 8
**Confidence:** 4

**Details Of Ethics Concerns:**

None.

**Main Review:**

On novelty:

I think if you asked any CRF practitioner to incorporate non-local hard constraints into a CRF, their first reaction would be to say, “I can hack it into the tag set.” And then they would have to hack the transition and emission feature functions to ignore the new information in the expanded tags to avoid parameter blow-up, and then the transition potential function to enforce the constraint. (I have done this.) They would eventually wind up with a one-off solution that looks very much like what the transformation described here would have handed them. Now the engineer is given another problem: to design a finite-state machine describing the output language, and to make it have as few edges as possible. I would argue that this is a step forward, and a worthwhile ML contribution.

Strengths:
- This is a useful addition to the CRF toolbox; it is a nice, clean formalism for adding regular language constraints, and by making the connection explicitly to regular languages and NFAs, it opens up the possibility of fruitful cross-pollination with formal language research.
- The paper is well-written and very easy to follow.
- The authors get out ahead of the inevitable question of the utility of their work in the face of more general work on learning weights for FSTs. I think Section 4.3 is strong, and gives a good argument why this contribution has value in the face of previous work.
The authors took the time to show that the technique helps even in the context of a state-of-the-art model.

Weaknesses:
- CRF’s quadratic dependence on tag set size, and the mapping from NFA edges to tags means that for many constraint sets, this solution will be infeasible.
- In general, I found 4.2 on tag-set minimization a little hard to follow. The advice amounts to, “minimize manually, and apply NFA minimization where applicable.” I think an example of an organic application of NFA minimization (plus a citation to the algorithm the authors have in mind!) would go a long way toward improving it.
- I didn’t find the proofs in Section 5 particularly compelling - they were easy to follow, but they extend almost trivially from definitions. In particular, the impact of training with constraints seems to (1) assume perfect minimization of the training objective and (2) ignore generalization error entirely. However, the synthetic data experiments do a good job compensating for this.
- The real-world experiment is not particularly convincing - it’s very good that the authors report statistical significance because the deltas in performance look very very small. I also think that it’s good that they attribute much of their strong performance to RoBERTa.
  - I think it would be informative to include the number constraint failures in the unconstrained model.
  - It would also be informative to include an unconstrained model that has the same computational concessions as the constrained model (removal of rare labels).


**Summary Of The Paper:**

This paper describes a transformation to add a hard regular language constraint to the output space of a linear chain CRF. Given a Non-deterministic Finite State Automaton (NFA) describing the CRF’s output space, their method maps the edges in that machine to a new CRF tag set, and wraps the potential function from the original CRF in a function that is aware of the compatibility of adjacent tags based on the edges they represent. They prove some basic properties about this transformation, and discuss its relation to learning weights in arbitrary FSTs. Finally, they provide experiments showing the technique’s application to synthetic data, as well as semantic role labeling, which has natural constraints such as uniqueness of core roles. They are able to show small but significant improvements over both an unconstrained ablation and the state-of-the-art on this dataset.

**Summary Of The Review:**

This paper formalizes and systematizes how to incorporate regular language constraints into CRF training and inference. This simplifies the incorporation of constraints, and makes it clear when they will become computationally infeasible. It also provides exciting hooks into formal language theory for future contributions. The experiments are not super-exciting, nor are the proofs, but the framework is a nice addition to CRFs overall.

---

> ### Author Response · Authors · 2021-11-11
> **Respose to Reviewer ephA**
>
> Thank you kindly for your feedback.  We are happy that you found our
> work to be a worthwhile contribution.
>
> We would like to adress a few of your concerns.
> > CRF’s quadratic dependence on tag set size, and the mapping from NFA edges to tags means that for many constraint sets, this solution will be infeasible.
>
> This is indeed a topic that
> deserves more discussion in our paper. We will expand
> section 4.2 to discuss performance more generally, discussing when
> our approach is intractable, when automaton minimization, manual or
> automatic, can make it tractable, and general performance tradeoffs
> between RegCCRFs and CRFs. We will make sure clarify the content in
> the process.
>
> >I didn’t find the proofs in Section 5 particularly compelling - they were easy to follow, but they extend almost trivially from definitions. In particular, the impact of training with constraints seems to (1) assume perfect minimization of the training objective and (2) ignore generalization error entirely. However, the synthetic data experiments do a good job compensating for this.
>
> We agree that they follow quite trivially from the definitions, but felt they needed to be stated explicitly to
> give proper context for the synthetic data experiments. To make
> space for further changes, we will possibly move the proofs
> themselves to an appendix, while keeping the theorem statements in the text.
> We agree that our assumptions are unrealistically restrictive -- as
> you note, we present the experiments to try to make up for these
> shortcomings.
>
> > * I think it would be informative to include the number constraint failures in the unconstrained model.
> > * It would also be informative to include an unconstrained model that has the same computational concessions as the constrained model (removal of rare labels).
>
> Both of your suggestions for
> improvement should be doable within the discussion period -- we can
> easily count the number of CRF predictions which violate our
> constraints, and we can use the same models we trained for
> constrained decoding, but decoded without constraints, to give a
> CRF-with-reduced-tagset baseline.  We will update our submission
> upon performing these experiments.

---

> > ### Comment · Reviewer_ephA · 2021-11-26
> > **Thanks**
> >
> > Thanks to the authors for their response. I'm glad to hear that you plan to include the two improvements I asked for in the next version.

---

### Official Review · Reviewer_82Jb · 2021-11-02

**Correctness:** 3
**Technical Novelty And Significance:** 3
**Empirical Novelty And Significance:** 3
**Recommendation:** 6
**Confidence:** 4

**Main Review:**

I liked this paper, but it took me a little while to overcome my initial hesitations about why one would want to do this. First, the application to BIO tagging is a good one, and it would help your presentation enormously to mention this application in the introduction. Second, readers may differ depending on their background, but I was initially confused about why one wouldn't want to just use finite transducers. This is explained adequately in Section 4.3, but I feel it is a little late. Perhaps the explanation can be left where it is, but summarized in the introduction.

I think the theorems in Section 5 are pretty intuitive, and if you are need of space, you could relegate the proof of Theorem 1 to an appendix. In Section 6, the experiments are interesting, but couldn't you go further and prove formally that a CRF is incapable of generating these particular string relations?

- I think there is a typo in the statement of Theorem 2; the two sides of the inequality are the same.

The method improves performance on semantic role labeling, but the improvements due to the proposed method (as opposed to using RoBERTa) are not dramatic. Nevertheless, they produce apparently the new SOTA on this task.

- On page 8, I'd like to see a clearer explanation of how the constraint language is constructed. There are a lot of magic numbers here that would benefit from explanation, and I certainly wouldn't be able to replicate the results from this explanation.

It had been claimed in Section 4.3 that direct comparison between the proposed method and neural-weighted finite transducers is possible. Maybe you meant direct theoretical comparison, but I definitely would have liked to see an experimental comparison between the two. The advantages claimed in Section 4.3 are legitimate, but I am not sure how much difference they make in practice.


**Summary Of The Paper:**

This paper describes CRFs that are constrained to generate tag sequences that belong to a given regular language (RegCCRF). This is useful, for example, in BIO tagging where tag sequences must be of the form O*(BI*O*)*. Since CRFs do not have hidden state, the constraint makes them more powerful. On the other hand, the claimed advantage of RegCCRFs over general finite transducers are (1) guarantee that the partition function converges, (2) finding the best path is the same as finding the best string, (3) the loss function is convex.


**Summary Of The Review:**

I like this paper and just think it needs some improved motivation in the introduction. An experimental comparison against neural-weighted finite transducers would strengthen the paper a lot by justifying the claims in Section 4.3.

---

> ### Author Response · Authors · 2021-11-11
> **Response to Reviewer 82Jb**
>
> Thank you for your helpful feedback.  We will take your suggestions
> into account in order to better motivate our introduction.
>
> >In Section 6, the experiments are interesting, but couldn't you go further and prove formally that a CRF is incapable of generating these particular string relations?
>
> Thanks for the suggestion! This proof is actually quite straightforward -- for any character pair, a CRF's marginals
> must be independent when conditioned on an intervening character (Markov Property),
> and we construct our distributions explicitly to break this
> independence assumption.  We will update the text to state this
> clearly.
>
> >I think there is a typo in the statement of Theorem 2; the two sides of the inequality are the same.
>
> You are correct about the typo in Theorem 2 -- the left side should
> of course be parameterized by $\theta_u$. We have will correct this shortly.
>
> >On page 8, I'd like to see a clearer explanation of how the constraint language is constructed. There are a lot of magic numbers here that would benefit from explanation, and I certainly wouldn't be able to replicate the results from this explanation.
>
>
> We
> will include such a description in the appendix, with pseudocode for
> the construction of such an automaton from a set of core roles,
> noncore roles, and continuation roles.
>
> >It had been claimed in Section 4.3 that direct comparison between the proposed method and neural-weighted finite transducers is possible. Maybe you meant direct theoretical comparison, but I definitely would have liked to see an experimental comparison between the two. The advantages claimed in Section 4.3 are legitimate, but I am not sure how much difference they make in practice.
>
> We did indeed mean theoretical comparisons, but you
> raise a very good point.  The main reason we did not include such a
> comparison is that it is not clear what a "canonical" neural-weighted
> FST would look like for this task. We saw two possibilities -- either we
> use the automaton we already generated, and learn edge weights for
> that automaton, or we construct a small highly-ambiguous automaton, and
> learn weights for that (a la Rastogi et al. (2016)).  In the former case, with no additional
> weight sharing, we would expect the high number of edges to make learning edge weights difficult.
> In the latter case, there is no obvious topology to choose -- for instance, we could chose that used in Rastogi et al. (2016),
> and exclude epsilon transitions (as our input and output sequences are always aligned), but the resulting
> model would simply be a CRF (albeit one with an unusual parameterization).

---

### Official Review · Reviewer_1EHx · 2021-11-02

**Correctness:** 4
**Technical Novelty And Significance:** 3
**Empirical Novelty And Significance:** 2
**Recommendation:** 6
**Confidence:** 4

**Main Review:**

I think the paper proposes a clever approach to deal with an interesting problem. The approach and constructions are simple and natural. The results in their few experiments also support the effectiveness of the approach. I was curious to see if they applied their method to a larger problem and if the approaches to optimize the computational costs made a significant difference.

They have an interesting discussion on the relationship between their approach and WFSTs. I understand that in contrast to WFSTs, this approach does not suffer from the issue of paths with unbounded lengths. But does it not arise from the need to induce a regular language before applying the approach which itself could be hard in some cases.

The paper is very well written and the arguments are very clearly presented. I appreciate the extra effort put in by the authors for the reproducibility and accessibility of their implementation.

I think authors could have explored more synthetic languages to solidify their claims. The synthetic experiments were on two very simple languages. I think a more systematic exploration even within the hierarchy of regular languages to test the limits of the approach would have been more insightful.

The improvement in the SRL task also seems to be incremental and the need to induce regular language for various practical structured prediction tasks could be difficult in certain cases.

**Summary Of The Paper:**

The paper proposes a modification over linear chain CRF models such that the output space of the model is constrained to be in a regular language. Linear Chain CRFs use Markov assumption where a given output only directly depends on its immediate neighbours which restricts the influence of distant ones. The assumption makes training tractable for the model but restricts its expressive power which could inhibit the performance on longer sequences. Some approaches have been previously proposed to relax the assumption but the authors claim they have certain drawbacks in terms of performance and expressive power.

The authors propose a new way to relax the Markov assumptions by constraining the output of a CRF to be in a regular language. They describe a simple way to construct such a constrained CRF when given an NFA by setting certain transition and emission probabilities to 0. They then discuss ways to make the algorithm more efficient by making use of equivalences classes and heuristics to minimize NFAs to minimize the size of tag sets. The authors also discuss the relationship between the constrained CRF model and weighted FSTs, and point certain distinctions in favour of the constrained CRF model.

Some previous constrained variants of CRFs are trained in the same way as standard CRFs, and then the constraints are enforced during decoding by setting certain output probabilities to 0. Unlike those approaches, the proposed approach could be trained and then used in a constrained manner. Training in a constrained fashion will directly minimize the NLL against the data distribution and achieve a better error, given output y \in regular language L.

The authors conduct two synthetic experiments to showcase that their model with constrained training is able to better capture non-local dependencies and data distributions compared to a model with constrained decoding. Additionally, they show slight improvements on a semantic role labeling task compared to baseline CRF models.

**Summary Of The Review:**

In summary, it seems like a good paper, with a simple and clever idea to improve a fundamental model in structured prediction. The arguments are made very clearly while presenting the idea but the paper lacks enough empirical evidence to back up the efficacy of the proposed idea.

---

> ### Author Response · Authors · 2021-11-11
> **Response to Reviewer 1EHx**
>
> Thank you for your feedback.
>
> > I think a more systematic exploration even within the hierarchy of regular languages to test the limits of the approach would have been more insightful.
>
> We agree that an investigation with
> more complicated artificial data could be insightful to better
> understand which data properties lead to significant improvements
> with constrained training.  We hope to do this in future work.
>
> >The improvement in the SRL task also seems to be incremental and the need to induce regular language for various practical structured prediction tasks could be difficult in certain cases.
>
> We would argue the opposite -- regular languages can provide a common framework for unifying many different structures. When approaching a new structured prediction task, defining a regular language for that structure will often be simpler than devising a new model architecture for that structure.

---

### Public Comment · ~Sean_Papay1 · 2023-09-11
**Minor update to account for a software bug**

Due to a software bug in our inference procedure, our initial publication reported evaluation scores which were lower than their true values. We have subsequently fixed this bug and performed a new evaluation and analysis. While none of our main findings are affected by this, some of the details in our analyses, such as the specifics of precision-recall tradeoffs, do look different. We have uploaded an updated version of this paper to arxiv [here](https://arxiv.org/abs/2106.07306), which reports accurate evaluation scores and which includes some updated analysis.

---

### Decision · Program_Chairs · 2022-01-20

**Decision:**

Accept (Poster)

**Comment:**

This paper does as it’s title suggests, it introduces an algorithm for constraining a CRF’s output space to correspond to a pre-specified regular language. The authors build upon a wealth of prior work aiming to enable CRFs to capture particular non-local dependencies and output constraints and present a coherent general algorithm to specify such constraints with a regular language. This is a clearly presented and well motivated contribution.

The reviewers predominantly agree that this work is clearly and rigorously presented and that the formalisation of constraints for CRFs through regular languages is a useful contribution for practitioners. One reviewer questioned the utility of constraining the output distribution at training time. In response the authors convincingly argue that unconstrained models will fail to learn the data generating distribution when non-local constraints exist in the data and have included a clear synthetic example of this in the paper.

The most significant weakness identified of this paper is the limited experimentation, consisting of one synthetic experiment and an application to semantic role labelling. The key motivation for formalising constraints on CRFs with regular languages is the argument that this allows model builders to use a familiar formalism across disparate tasks rather than producing bespoke solutions for each. As such it would be informative when assessing the contribution of this work to see a number of practical examples of task output spaces formalised as regular languages such that we can form an intuition for how natural this representation is for more than one task, while also shedding light on the ease, or otherwise, of the crucial processing of minimising the representation to maximise efficiency.

While the application to a broader range of tasks would definitely strengthen this paper, in its current form it provides a useful formalism that will be of interest to those working in structured learning and as such is a contribution worthy of publication.